# Multimodal, Personalized Treatment of Pineal Region Tumors in Adulthood—A Single Center Study

**DOI:** 10.3390/jcm15010248

**Published:** 2025-12-29

**Authors:** Tamás Mezei, János Báskay, Péter Pollner, Lukács Németh, Balázs Markia, Gábor Nagy, András Bajcsay, László Sipos

**Affiliations:** 1Department of Neurosurgery and Neurointervention, Semmelweis University, 1145 Budapest, Hungary; 2Data-Driven Health Division of National Laboratory for Health Security, Health Services Management Training Centre, Semmelweis University, 1125 Budapest, Hungary; 3Department of Biological Physics, Eötvös Loránd University, 1117 Budapest, Hungary; 4National Institute of Oncology, 1122 Budapest, Hungary

**Keywords:** pineal tumors, endoscopic intervention, radiotherapy, radiosurgery, personalized therapy

## Abstract

**Background:** Tumors of the pineal region account for less than 1% of supratentorial neoplasms in adults and represent a distinct neuro-oncological challenge. Their management requires a multidisciplinary and multimodal approach. Traditionally, direct surgical resection was considered the primary treatment modality. Recent advances in minimally invasive techniques and onco-radiotherapy have paved the way for safer and more personalized treatment strategies, in line with the principles of precision medicine. This study aims to present our institutional approach, which relies on a combination of endoscopic and radiotherapy-based techniques. **Methods:** A retrospective, single-center clinical study was conducted involving 28 adult patients who underwent endoscopic third ventriculostomy and biopsy of a pineal region tumor between January 2014 and March 2025. Descriptive statistics, permutation tests with bootstrap-derived confidence intervals, Fisher’s exact test, and Kaplan–Meier survival analysis were applied for data evaluation. **Results:** Endoscopic intervention resulted in clinical improvement in 78% of cases. A significant increase in performance status was observed in the postoperative period (<0.001) compared to preoperative results. Radiotherapy contributed to either tumor regression or disease stabilization. **Conclusions:** Based on our findings, the combination of endoscopic intervention and personalized radiotherapy represents a safe and effective treatment strategy, offering a compelling alternative to direct surgical resection, which is reserved as a second-line treatment.

## 1. Introduction

Because of their anatomic location, tumors of the pineal region can present with a wide spectrum of clinical symptoms. However, almost all patients exhibit signs of disturbed cerebrospinal fluid (CSF) circulation and associated symptoms. Other possible manifestations include gaze palsy, endocrinological abnormalities (e.g., precocious puberty), and spinal symptoms in cases of drop metastases [1].

The surgical management of tumors located in the pineal region is an exceptionally complex and challenging task due to its deep, axial location and proximity to eloquent structures including the third ventricle, splenium of the corpus callosum, thalamus, lamina tecti, and deep veins. The treatment is further complicated by the extreme rarity of these tumors, accounting for only 3–8% of pediatric central nervous system tumors and less than 1% of adult supratentorial tumors [2,3]. Moreover, the histological heterogeneity of tumors arising in this region adds to the complexity, with germinomas and astrocytomas predominating in children, while meningiomas and gliomas are more frequent in adults.

Owing to the rarity of the disease, large cohort studies providing definitive treatment guidelines are lacking. According to the literature, surgical resection remains the cornerstone of treatment for benign tumors. However, this notion is increasingly being challenged by advancements in minimally invasive techniques and radiotherapy modalities [4,5]. In cases of malignant morphology, the currently recommended management consists of endoscopic CSF diversion and biopsy, followed by adjuvant oncological therapy [6,7].

The mortality rate associated with surgical intervention for pineal region tumors is below 3%; however, the risk of postoperative morbidity may reach up to 20% [4,8]. In contrast, combined endoscopic–radiotherapeutic management is associated with a morbidity rate of less than 3% [9].

Over the years, the treatment of pineal region tumors has undergone significant advancement, and the adoption of multimodal approaches enables patients to access safer therapeutic options with better preservation of quality of life.

The aim of our study is to present the clinical practice developed at our institution, which is based on novel minimally invasive endoscopic techniques and radiotherapy (Figure 1 and Figure 2).

## 2. Methods

We conducted a single-center retrospective clinical study analyzing patient data from the Department of Neurosurgery and Neurointervention, Semmelweis University. Our aim was to identify the adult patient population—defined as individuals aged 18 years or older at the time of surgery—who underwent endoscopic third ventriculostomy (ETV) and biopsy for tumors located in the pineal region causing triventricular hydrocephalus between January 2014 and March 2025, to analyze their clinical characteristics, and perform a statistical evaluation. Radiotherapy or radiosurgery was performed at the National Institute of Oncology or at the Uzsoki Hospital in Budapest.

During the time interval described above, a total of 28 patients were treated and followed up at our department.

For each patient, comprehensive data were collected covering the preoperative, intraoperative, and postoperative periods. The parameters included sex, age, presenting symptoms (signs of increased intracranial pressure, symptoms of chronic hydrocephalus, and signs of local/tectal lesions), symptom duration at admission, pre- and postoperative Karnofsky Performance Status (KPS), ECOG score, date of surgery, need for reoperation and its type (re-ETV and/or VPS implantation and/or transcranial tumor removal), detailed histopathological diagnosis, specifics of adjuvant radiotherapy, documentation of clinical changes during outpatient follow-up, timing of these changes, and date of death. Mortality data were obtained from the National Cancer Registry operated by the National Institute of Oncology.

### Statistical Analysis

Descriptive statistics were performed to summarize patient data.

Fisher’s exact test was applied to determine the presence of statistically significant associations between two categorical variables (e.g., whether there is significant correlation between symptom onset and histological categories). This test requires the construction of a contingency table for the two variables. The *p* value indicates the probability, under the null hypothesis, of obtaining data as extreme as the results observed.

Numerical data were compared using permutation tests (10,000 resamples, two-sided) to assess whether observations at different timepoints demonstrated statistically significant differences. All possible pairwise comparisons were made, testing the null hypothesis of equal means for each. Bootstrap resampling (10,000 resamples, percentile method) was used to estimate 95% confidence intervals around mean differences. These nonparametric methods were employed to account for the small sample size and provide robust estimates without assuming normality of the underlying distributions.

Progression-free survival (PFS) and overall survival (OS) were also analyzed using the Kaplan–Meier method.

A significance threshold of 5% (*p* < 0.05) was applied for all statistical tests. Analyses were performed using R statistical software (version 4.1.0). We emphasize that, due to the small sample size, our findings should be interpreted with appropriate caution.

Limitations:

Our study has several important limitations that may affect the reliability and generalizability:

Retrospective nature of the study.

Single-center design

Small sample size of 28 patients with varied pathologies, leading to class imbalance.

Follow-up data are incomplete (e.g., loss to follow-up due to poor performance).

Radiotherapy was administered in multiple centers, which further increases the heterogeneity of the treatments

## 3. Results

### 3.1. Descriptive Statistics

The patient cohort included 14 females (50%) and 14 males (50%). Two-thirds of the patients were between 30 and 60 years of age, with the youngest patient aged 29 years, the oldest 74 years, and the mean age being 50 years.

Three major groups of presenting symptoms were distinguished. The first group was chronic CSF circulation disorder, most commonly presenting as Hakim’s triad (gait disturbance, dementia, urinary incontinence), observed in 42.9% of patients. The second one consisted of functional deficits resulting from local compression of the tectal region and manifested as gaze and ocular motor disturbances (e.g., Parinaud’s syndrome, i.e., vertical gaze palsy), affecting 17.9% of patients. The third group comprised signs of increased intracranial pressure such as headache, nausea, vomiting, and altered consciousness, most frequently associated with aqueductal stenosis causing triventricular hydrocephalus. This affected 71.4% of patients in varying degrees of severity. More than half of the patients developed symptoms within 3 months from an asymptomatic state (<1 month: 35.7%; 1–3 months: 17.9%). In 4 cases (14.3%), symptoms persisted for over one year prior to the diagnosis of pineal tumor.

Pathological findings revealed that the largest proportion of tumors originated from the pineal parenchyma (pineocytoma: 17.9%, papillary tumor of the pineal region: 10.7%, pineal parenchymal tumor with intermediate differentiation: 14.3%), followed by metastases (10.7%) and other extraparenchymal tumors (e.g., meningioma: 7.1% or solitary fibrous tumor: 7.1%). The detailed distribution of histological results in our patient cohort is shown in Figure 3.

Our results indicate that 78.6% of cases showed clinical improvement following endoscopic intervention, with no new neurological deficits attributable to the surgery. A second operation was required due to liquor passage disorder in six cases, when the stoma created during fenestration did not provide adequate CSF passage. In these instances, ventriculoperitoneal shunt implantation provided a permanent solution in five cases. When a longer interval had passed after the initial endoscopic third ventriculostomy (ETV), repeat endoscopic intervention was also attempted. Additionally, in cases of direct brainstem compression caused by the tumor, urgent craniotomy was performed to achieve partial or complete tumor resection. The types and percentages of reoperations following ETV are presented in Figure 4.

### 3.2. Analysis of Clinical Parameters

To assess possible associations between clinical parameters, Fisher’s exact test was applied. The null hypothesis assumed independence between two categorical variables (e.g., presence of local symptoms and symptom onset within one month), i.e., odds ratio (OR) = 1. If the OR significantly deviated from 1 (*p* < 0.05), the null hypothesis was rejected.

No statistically significant associations were found when comparing presenting symptoms and timing of symptom onset in contingency tables; however, OR values deviating from 1 were observed in several instances:

local symptoms—symptom onset within <1 month: OR = 3.266 (95% CI: 0.303–47.446), *p* = 0.315;

signs of elevated intracranial pressure (ICP)—symptom onset within <1 month: OR = 5.421 (95% CI: 0.526–285.779), *p* = 0.194.

Due to the wide confidence intervals (primarily attributable to the small sample size) these tests are considered inconclusive. Nevertheless, our findings suggest that local symptoms and signs of elevated intracranial pressure tend to be associated with early symptom onset, likely reflecting the presence of more severe clinical states or larger tumor burden.

When analyzing the relationship between preoperative symptoms and performance status, similarly inconclusive associations were observed between local symptoms and low KPS scores (0–40) (OR = 5.030 (95% CI: 0.056–448.718), *p* = 0.331). A statistically significant association was found between the presence of chronic hydrocephalus and moderate KPS scores (50–70) (OR = 17.963 (95% CI: 2.255–257.679), *p* = 0.001).

The relationship between histopathological categories and clinical features was also investigated. However, due to the large number of subcategories combined with low case counts per group, resulting in many contingency table cells with one or zero cases, precluded meaningful statistical analysis.

### 3.3. Survival Analysis and Investigation of Postoperative Radiotherapy Effectiveness

Table 1 presents the distribution of postoperative radiotherapy modalities across histopathological categories, Table 2 shows the corresponding median PFS and OS values. For patients without disease progression, the last outpatient follow-up date was used as the censoring point in the PFS analysis. The Kaplan–Meier curve of the cohort’s PFS is shown in Figure 5. For the OS analysis, censoring was applied at 1, 2, and 5 years postoperatively. Patients who had not yet reached these follow-up intervals were excluded from the OS analysis. Seven patients died following the interventions, while 19 patients are currently under follow-up.

Table 2 summarizes the therapeutic response to conformal radiotherapy and stereotactic radiosurgery. Although no statistically significant difference was found, the elevated OR values suggest a trend: fractionated conformal radiotherapy appeared to be associated with more frequent tumor regression (OR = 3.823, *p* = 0.194), while stereotactic radiosurgery was more likely to result in radiologically stable disease (OR = 3.178, *p* = 0.207). No radiological progression was observed in any of the followed patients. (These findings should be interpreted with caution, as histopathologically homogeneous groups could not be established for meaningful comparison.)

Table 3 summarizes the therapeutic response to conformal radiotherapy and stereotactic radiosurgery. Although no statistically significant difference was found, the elevated OR values suggest a trend: fractionated conformal radiotherapy appeared to be associated with more frequent tumor regression (OR = 3.823 (95% CI: 0.534–31.808), *p* = 0.194), while stereotactic radiosurgery was more likely to result in radiologically stable disease (OR = 3.178 (95% CI: 0.403–28.955), *p* = 0.207). No radiological progression was observed in any of the followed patients. (These findings should be interpreted with caution, as histopathologically homogeneous groups could not be established for meaningful comparison.)

As the final step in our analysis, we compared KPS values recorded at different timepoints during treatment using permutation tests with bootstrap-derived confidence intervals. The results are presented in Table 4. We found statistically significant differences between preoperative KPS and postoperative KPS (mean difference: 11.1 (95% CI: 1.4–20.4), *p* = 0.037), as well as between preoperative KPS and first postoperative outpatient follow-up (mean difference: 20.5 (95% CI: 13.5–28.0), *p* < 0.001) and the first outpatient visit following radiotherapy (mean difference: 20.0 (95% CI: 11.0–28.6), *p* < 0.001). These results suggest a consistent and significant improvement in patient functional status after treatment.

## 4. Discussion

Tumors of the pineal region account for less than 1% of adult brain neoplasms and represent a highly heterogeneous group from a histopathological perspective. They are typically classified into germ cell tumors, pineal parenchymal tumors, neuroepithelial tumors, and other lesions (e.g., extraparenchymal lesions) [10]. According to Lu et al. [11], pineal parenchymal tumors and germ cell tumors together comprise approximately 70% of all neoplasms in this region. In our cohort, this proportion was notably lower, with these two categories accounting for less than 50% of cases. Compared to the literature, we observed a higher incidence of neuroepithelial and extraparenchymal lesions. While metastases to the pineal region are considered extremely rare in the literature [12], they represented 10.7% of cases in our series.

Despite the special localization and the varying histological diagnoses, the clinical presentation tends to standardize management approach. According to the findings of Yamini et al. [13], the incidence of obstructive hydrocephalus can reach up to 90%. Elevated intracranial pressure caused by aqueductal stenosis is potentially life-threatening, which underscores the critical importance of surgical intervention in this region.

In the management of pineal region tumors, the primary goal is to relieve CSF circulation disturbances and reduce ICP. Currently, ETV is favored over VPS implantation [14]. One major advantage of ETV is that it allows for biopsy under direct visual control, offering high sensitivity and a low complication rate [6,15,16]. According to the literature, stoma dysfunction occurs in approximately 15% of cases [4], which aligns with our findings: 17.9% of patients in our cohort required permanent shunt placement following endoscopic fenestration.

Direct surgical management of pineal tumors is recommended only by neurosurgeons with appropriate experience, preferably performed in specialized centers. The choice of surgical approach is primarily determined by the tumor’s relationship to the vein of Galen. Available approaches include infratentorial supracerebellar, interhemispheric transcallosal, occipital transtentorial, and transcortical transventricular routes [8]. In a retrospective study by Shepard et al. [17], outcomes of 68 patients undergoing surgery were analyzed. Postoperative complications occurred in 45.6% of cases, including hydrocephalus requiring further surgery in 14.7% and focal neurological deficits in 10.3%. The 30-day mortality rate was 5.9%. Macroscopic total resection was achieved in 52.9% of patients.

Radiotherapy remains a cornerstone of multimodal treatment for pineal region tumors, particularly in cases of germinomas, non-germinomatous germ cell tumors, and high-grade parenchymal neoplasms [18,19,20]. Advances in radiation techniques have enabled more conformal dose distribution and improved target volume coverage, while minimizing exposure to surrounding normal tissues. In our cohort, radiotherapy was administered at the National Institute of Oncology and at the Uzsoki Hospital. Stereotactic radiosurgery using the CyberKnife system (single or multi-fraction regimens) was performed in 11 patients, and conformal fractionated radiotherapy in 9 cases. In patients with metastatic disease, whole-brain radiotherapy (10 × 3 Gy) was employed. Glioblastoma patients received temozolomide chemoradiation according to the Stupp protocol [21]. Patients received oncological treatment in accordance with current guidelines and tailored to their primary tumor type. We evaluated the relationship between radiotherapeutic effects and treatment modalities. Although not statistically significant, we observed elevated odds ratios suggesting improved tumor control in terms of stability (OR = 3.178 (95% CI: 0.403–28.955)) and radiological regression (OR = 3.823 (95% CI: 0.534–31.808)). The combined endoscopic and radiotherapy approach resulted in a statistically significant improvement in post-treatment performance status compared to preoperative values (*p* < 0.001). Notably, the most significant improvement in KPS occurred immediately after surgery (*p* = 0.0310). No statistically significant difference was found between postoperative and post-radiotherapy KPS values (*p* = 0.130). These findings suggest that the resolution of hydrocephalus accounts for an immediate improvement in performance status, while radiotherapy plays a key role in maintaining this benefit over time through effective tumor control.

Our results align with the literature, indicating that radiotherapy and radiosurgery can serve as alternatives to direct surgical treatment [22]. Zeyna et al. [23] treated histologically unverified pineal region tumors using stereotactic radiosurgery. Sixty percent of their patients responded to the treatment, with complete remission achieved in 10%. Following therapy, the patients’ KPS improved by 8 points on average.

We would like to emphasize that outcome and long-term survival are primarily determined by histological diagnosis and tumor behavior. Due to the small sample size in our survival analysis, we were unable to draw meaningful conclusions.

Based on these findings, minimally invasive endoscopic third ventriculostomy combined with biopsy and subsequent radiotherapy provides a durable solution for patients and maintains an appropriate quality of life. Given its low burden, this approach is also applicable to older patients and those with significant comorbidities. If adequate tumor control is not achieved with the above combination therapy, we recommend direct surgical resection as a second-line treatment in properly selected patients. Figure 6 presents our institutional therapeutic algorithm recommendation for the treatment of pineal region tumors.

## 5. Conclusions

The complexity and challenges inherent to the treatment of pineal region tumors necessitate a multidisciplinary approach that maximizes the benefits of each modality while minimizing potential risks and complications. The combination of minimally invasive endoscopic surgery and adjuvant radiotherapy represents a promising, compromise-free therapeutic alternative to direct surgical management for both benign and malignant lesions in the pineal region. With a lower rate of perioperative complications, our results demonstrate that patients’ quality of life can be significantly improved through endoscopic intervention, and the subsequent radiotherapy effectively maintains this benefit over time.

Based on experience at our institution, we recommend, as first-line treatment, initiating endoscopic cerebrospinal fluid diversion alongside biopsy for tumors around the pineal gland (utilizing neuronavigation assistance in the absence of hydrocephalus), followed by the selection of the most appropriate radiotherapy modality.

Direct surgical resection is reserved as a second-line option, indicated in cases of rapid disease progression, significant local mass effect, hydrocephalus unamenable to endoscopic treatment, radioresistant tumor types, or progression detected during active therapy.

## Figures and Tables

**Figure 1 jcm-15-00248-f001:**
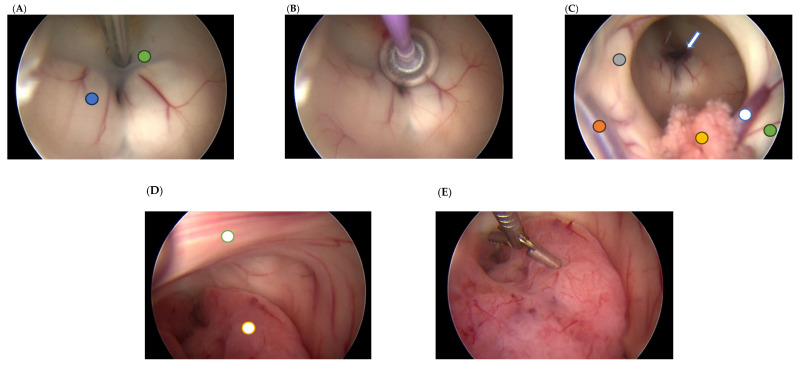
Intraoperative images of endoscopic third ventriculostomy and biopsy. (**A**) Fenestration of the third ventricular floor and the Liliequist membrane using a dedicated bipolar instrument for ventriculoscopy (green circle—floor of the third ventricle; blue circle—left mammillary body). (**B**) Enlargement of the created opening using a Fogarty catheter. (**C**) Patent stoma viewed from the right foramen of Monro (hollow blue arrow) (orange circle—vein of septum pellucidum; hollow blue circle—vein of thalamostriate; yellow circle—choroid plexus; green circle—head-body junction of the caudate nucleus; grey circle—free part of the fornix column). (**D**) Posterior view of the third ventricle showing a reddish tumor occupying the aqueduct (hollow yellow circle) and the interthalamic adhesion (hollow green circle). (**E**) Tissue sampling using a micro-rongeur.

**Figure 2 jcm-15-00248-f002:**
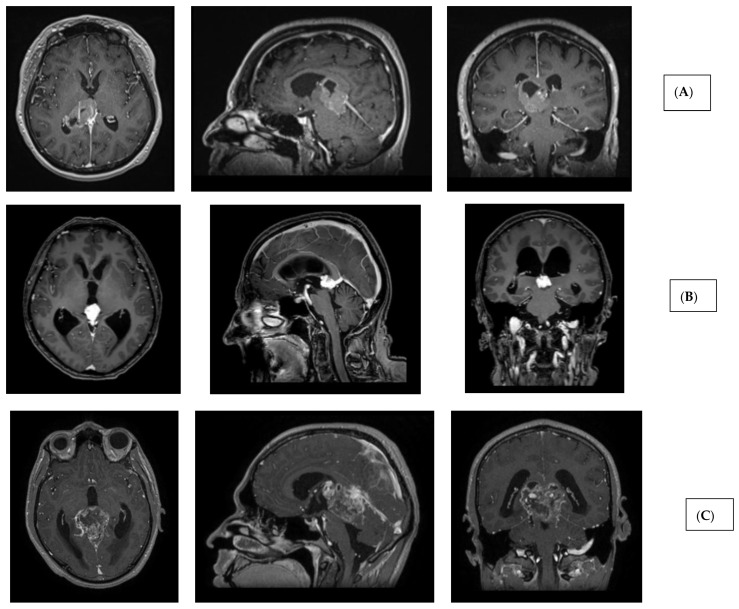
Imaging characteristics of pineal region tumors and post-therapeutic changes. (**A**) Pineal parenchymal tumor (histology: pineal parenchymal tumor with intermediate differentiation, CNS WHO grade 2/3) (T1 contrast-enhanced MRI sequences; axial, sagittal, and coronal views). (**B**) Extraparenchymal tumor in the pineal region (histology: solitary fibrous tumor, CNS WHO grade 2) (T1 contrast-enhanced MRI sequences; axial, sagittal, and coronal views). (**C**) Glial (neuroepithelial) tumor in the pineal region (histology: glioblastoma, CNS WHO grade 4) (T1 contrast-enhanced MRI sequences; axial, sagittal, and coronal views). (**D**) Germ cell tumor of the pineal region (histology: germinoma). Left: preoperative image (T1 contrast-enhanced MRI, axial slice). Middle: post-radiotherapy control showing complete radiological remission (T1 contrast-enhanced MRI, axial slice). Right: post-radiotherapy follow-up (T2 SPACE MRI, sagittal slice), demonstrating patent third ventricular stoma (hollow blue arrow) and aqueduct with prominent CSF flow artifacts (hollow green arrow). (**E**) Extraparenchymal tumor in the pineal region—immediate postoperative T2 SPACE images—left sagittal image shows mild CSF flow artifact (blue arrow), right image shows compressed, nonfunctional aqueduct. (**F**) Extraparenchymal tumor in the pineal region—T2 SPACE images one year after radiotherapy—left sagittal image shows no CSF flow artifact (stoma closure), right image shows reopened, functional aqueduct (orange arrow) (tumor regression following radiotherapy).

**Figure 3 jcm-15-00248-f003:**
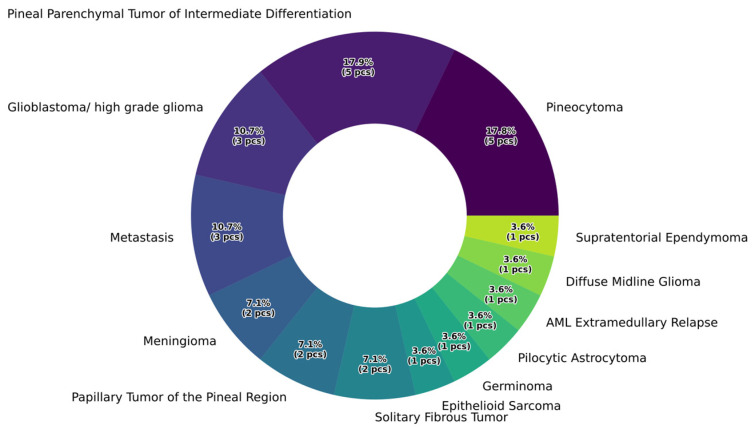
Percentage distribution of histopathological categories.

**Figure 4 jcm-15-00248-f004:**
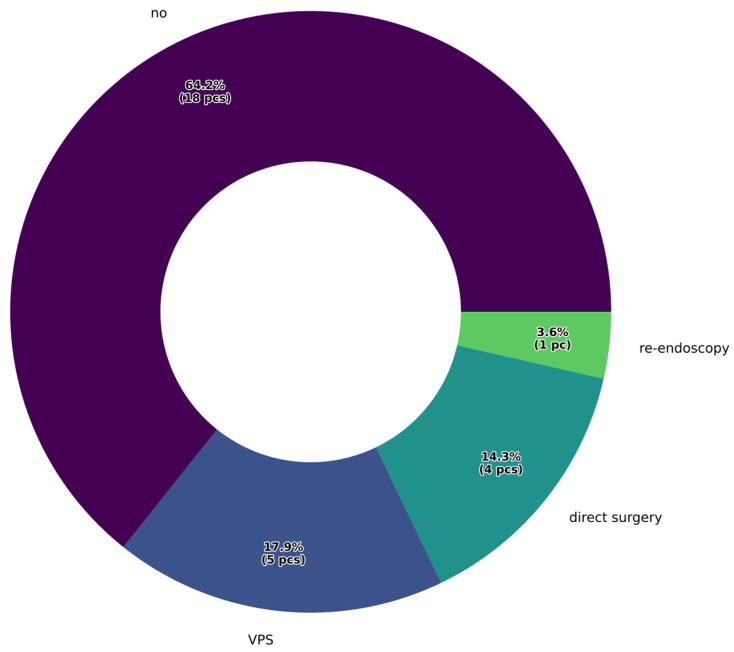
Percentage distribution of reoperations following endoscopic surgery.

**Figure 5 jcm-15-00248-f005:**
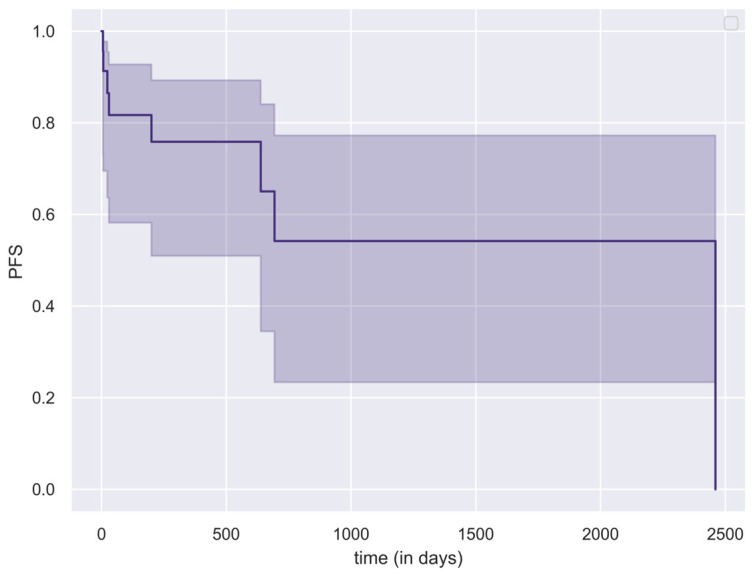
Kaplan–Meier curve (with 95% CI intervals) illustrating progression-free survival (PFS) of the study population.

**Figure 6 jcm-15-00248-f006:**
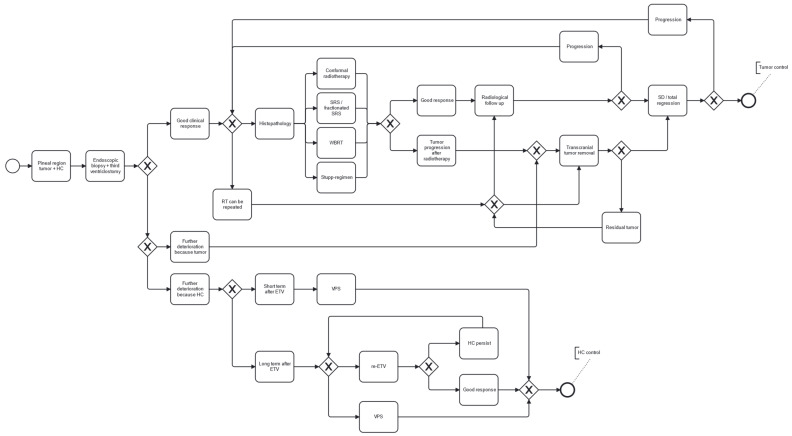
Therapeutic algorithm recommendation for the treatment of pineal region tumors.

**Table 1 jcm-15-00248-t001:** The detailed presentation of postoperative radiotherapy according to histopathological categories. For five patients, no information is available regarding radiotherapy or the postoperative period (loss to follow-up).

Patient ID	Histology	Modality/Technique	RT Equipment	Image Guidance	RT Planning System	Target Delineation	CTV-PTV Margins	Dose/Fractionation	Organs At Risk Constraints	CTCAE Grading
1	Specimen with limited evaluability (high-grade glioma)	Conformal RT	LINAC Unique	Megavoltage image guidance	Eclipse	Radiation Oncologist + Neurosurgeon	15 mm	60 Gy/30 fr	according to AAPM Task Group 101	-
2	Germinoma	Conformal RT	LINAC Siemens Artiste	Megavoltage CBCT	Pinnacle		2 + 3 mm	36 Gy/20 fr + pineal boost 18 Gy/9 fr		Gr.2
3	Metastasis (pulmonary adeno cc.)	WBRT	LINAC Siemens Primus	Kilovoltage CBCT	Eclipse		2 mm	30 Gy/10 fr		-
4	Pineocytoma (WHO grade 1)	SRS	LINAC Varian TrueBeam	Kilovoltage CBCT	Eclipse		2 mm	15 Gy/1 fr		-
5	Metastasis (breast cc.)	WBRT	LINAC Siemens Primus	Kilovoltage CBCT	Eclipse		2 mm	30 Gy/10 fr		-
6	Epithelioid sarcoma (proximal type)/adult rhabdoid tumor	NA					NA			NA
7	Pilocytic astrocytoma (WHO grade 1) with BRAF-KIAA1549 gene fusion	SRS	LINAC Varian TrueBeam	Kilovoltage CBCT	Eclipse		2 mm	14 Gy/1 fr		-
8	AML Extramedullary Relapse	WBRT + Conformal boost	LINAC Varian VitalBeam	Kilovoltage CBCT	Eclipse		2 + 3 mm	30.6 Gy/17 fr + pineal boost 5.4 Gy/3 fr		Gr.1
9	Pineocytoma (WHO grade 1)	fSRT	CyberKnife	X-Ray image feedback + Robotic arm	Precision		1 mm	30 Gy/5 fr		-
10	Pineocytoma (WHO grade 1)	fSRT	CyberKnife	X-Ray image feedback + Robotic arm	Precision		1 mm	30 Gy/5 fr		-
11	Metastasis (SCLC)	WBRT	LINAC Siemens Primus	Kilovoltage CBCT	Eclipse		2 mm	30 Gy/10 fr		-
12	Meningioma (WHO grade 1)	fSRT	CyberKnife	X-Ray image feedback + Robotic arm	Precision		1 mm	30 Gy/5 fr		Gr.1
13	Papillary tumor of the pineal region (WHO grade 2–3)	NA					NA			NA
14	Meningioma (WHO grade 2)	fSRT	CyberKnife	X-Ray image feedback + Robotic arm	Precision		1 mm	30 Gy/5 fr		-
15	Pineocytoma (WHO grade 1)	Conformal RT	LINAC Truebeam	Kilovoltage CBCT	Eclipse		4 + 5 mm	54 Gy/27 fr		-
16	Pineocytoma (WHO grade 1)	NA					NA			NA
17	Diffuse midline glioma (WHO grade 4) with H3 K27M mutation and ATRX loss	VMAT	LINAC Truebeam	Kilovoltage CBCT	Eclipse		15 + 4 mm	46 Gy/23 fr		-
18	Pineal parenchymal tumor (WHO grade 2–3)	WBRT + Conformal boost	LINAC Varian VitalBeam	Kilovoltage CBCT	Eclipse		2 + 3 mm	30 Gy/15 fr + pineal boost 25 Gy/10 fr		Gr.1
19	Pineal parenchymal tumor (WHO grade 2–3)	fSRT	CyberKnife	X-Ray image feedback + Robotic arm	Precision		1 mm	25 Gy/5 fr		-
20	Solitary fibrous tumor (WHO grade 1)	fSRT	CyberKnife	X-Ray image feedback + Robotic arm	Precision		1 mm	22.5/5 fr		-
21	Pineal parenchymal tumor (WHO grade 2–3)	fSRT	CyberKnife	X-Ray image feedback + Robotic arm	Precision		1 mm	25 Gy/5 fr		-
22	Supratentorial ependymoma, NOS (WHO grade 3), with pTERT mutation.	fSRT	CyberKnife	X-Ray image feedback + Robotic arm	Precision		2 mm	32.5 Gy/5 fr		-
23	Glioblastoma, NOS (WHO grade 4), with ATRX loss.	NA					NA			NA
24	Pineal parenchymal tumor (WHO grade 2–3)	fSRT	CyberKnife	X-Ray image feedback + Robotic arm	Precision		1 mm	25 Gy/5 fr		-
25	Papillary tumor of the pineal region (WHO grade 2–3)	VMAT	Halcyon (LINAC)	Kilovoltage CBCT	Eclipse		5 mm	54 Gy/27 fr		-
26	Glioblastoma with chromosome 7 alteration (WHO grade 4).	VMAT	Halcyon (LINAC)	Kilovoltage CBCT	Eclipse		6 + 3 mm	30.6 Gy/17 fr + pineal boost 24 Gy/12 fr + pineal boost 4 Gy/2 fr		-
27	Solitary fibrous tumor (WHO grade 1)	NA					NA			NA
28	Pineal parenchymal tumor (WHO grade 2–3)	Conformal RT	LINAC Varian TrueBeam	Kilovoltage CBCT	Eclipse		2 + 3 mm	36 Gy/20 fr + Pineal boost 18 Gy/9 fr		Gr.1

**Table 2 jcm-15-00248-t002:** Survival analysis with median overall survival (OS) and progression-free survival (PFS) values corresponding to each histopathological category.

Histopathology	Cases [Count]	Median PFS [Days]	Median OS (1y) [Days]	Median OS (2y) [Days]	Median OS (5y) [Days]
AML Extramedullary Relapse	1	-	365	730	1826
Diffuse Midline Glioma	1	93	270	270	270
Epithelioid Sarcoma	1	-	271	271	271
Germinoma	1	30	>365	>730	>1826
Glioblastoma/high grade glioma	3	14	>365	>730	1826
Meningioma	2	1267	>365	>730	>1826
Metastasis	3	638	365	690	690
Papillary Tumor of the Pineal Region	2	177	365	>730	1826
Pilocytic Astrocytoma	1	200	365	730	1826
Pineal Parenchymal Tumor of Intermediate Differentiation	5	578	365	730	788
Pineocytoma	5	758	365	730	1826
Solitary Fibrous Tumor	2	137	326	510	288
Supratentorial Ependymoma	1	98	>365	>730	-

**Table 3 jcm-15-00248-t003:** Results of Fisher’s exact test assessing the therapeutic effects of different radiotherapy modalities.

	Changes on Control MRI	OR	OR 95% CI	*p*
**Conformal Radiotherapy**	regression	3.823	[0.534, 31.808]	0.194
	unchanged	0.313	[0.006, 3.628]	0.380
	progression	0.000	[0.0, 73.583]	1.000
**Stereotactic Radiosurgery**	regression	1.698	[0.243, 11.635]	0.677
	unchanged	3.178	[0.403, 28.955]	0.207
	progression	inf	[0.046, inf]	0.357

**Table 4 jcm-15-00248-t004:** Pairwise comparisons of Karnofsky Performance Status (KPS) at different treatment timepoints. Mean differences with 95% bootstrap confidence intervals and permutation test *p*-values are reported.

	Preop KPS	Postop KPS	First Postoperative Outpatient Control KPS	KPS at First Outpatient Control Following Radiotherapy
**Postop KPS**	11.1 [1.4, 20.4], *p* = 0.037	-	-	-
**First postoperative outpatient control KPS**	20.5 [13.5, 28.0], *p* < 0.001	18.3 [8.3, 27.8], *p* < 0.001	-	-
**KPS at first outpatient control following radiotherapy**	20.0 [11.0, 28.6], *p* < 0.001	7.5 [−1.5, 17.5], *p* = 0.164	7.2 [−4.4, 18.9], *p* = 0.280	-
**KPS at last outpatient Visit**	7.1 [−3.3, 17.6], *p* = 0.224	2.5 [−2.5, 7.5], *p* = 0.474	0.0 [−8.0, 6.5], *p* = 1.000	−3.5 [−12.4, 2.9], *p* = 0.553

## Data Availability

The datasets used and/or analyzed in this study are available from the corresponding author upon reasonable request.

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
