# Peer review of "Multimodal, Personalized Treatment of Pineal Region Tumors in Adulthood—A Single Center Study"

_jcm, 2025, doi:10.3390/jcm15010248_

Round 1

Reviewer 1 Report

Comments and Suggestions for Authors

This study provides valuable insight into the multimodal management of adult pineal region tumors, yet several areas could be refined to strengthen scientific soundness.

1) The retrospective, single-center design should be more critically addressed, noting risks of selection bias and limited generalizability. Clear inclusion and exclusion criteria, along with a patient flow diagram, would improve methodological transparency. Comparison with external cohorts could contextualize representativeness.

2) Given the small cohort, statistical power is limited. Confidence intervals should accompany p-values, and nonparametric or bootstrap methods are preferable. Exploratory multivariate analyses (e.g., Cox regression) could identify independent prognostic factors.

3) Detailed description of dose, fractionation, target delineation, and planning technique is needed for reproducibility. Reporting organ-at-risk constraints, image guidance, and toxicity (CTCAE grading) would strengthen evaluation of safety and efficacy.

4) The analysis should align with the WHO 2021 CNS classification and incorporate molecular markers (e.g., IDH mutation, MGMT, TERT). Including such data would enhance biological interpretation and comparability with current neuro-oncology standards.

Author Response

We would like to thank the reviewer for their work, valuable comments, and constructive guidance.

This study provides valuable insight into the multimodal management of adult pineal region tumors, yet several areas could be refined to strengthen scientific soundness.

  • The retrospective, single-center design should be more critically addressed, noting risks of selection bias and limited generalizability. Clear inclusion and exclusion criteria, along with a patient flow diagram, would improve methodological transparency. Comparison with external cohorts could contextualize representativeness.

We revised the manuscript to emphasize that it presents our specific clinical practice for a rare disease. This involved minor rephrasing across several sections and the addition of a dedicated “Limitations” subsection to the “Methods” chapter, explicitly detailing the scope of our treatment algorithm.

Given the rarity of the condition, the available literature predominantly consists of single-center studies with small patient cohorts (fewer than 100 cases), as well as systematic reviews—some of which we have cited. We do not believe that providing a detailed breakdown of patient numbers from each referenced study would enhance the clarity of our manuscript or improve the delivery of our message.

Examples:

Ref. 7 – Morgenstern PF, Osbun N, Schwartz TH, Greenfield JP, Tsiouris AJ, Souweidane MM (2011) Pineal region tumors: an optimal approach for simultaneous endoscopic third ventriculostomy and biopsy. Neurosurg Focus 30: E3.

Patient number: 15

Ref. 14 – Cipri S, Gangemi A, Cafarelli F, Messina G, Iacopino P, Al Sayyad S, Capua A, Comi M, Musitano A (2005) Neuroendoscopic management of hydrocephalus secondary to midline and pineal lesions. J Neurosurg Sci 49: 97–106.

Patient number: 14

Ref. 23 – Zeynal M, Karaaslan B, Dagli O, Borcek A, Kurt G, Kadioglu HH, Emmez OH (2023) Stereotactic radiosurgery for tumors of the pineal region: A single-center experience. Medicine (Baltimore) 102: e34005.

Patient number: 25

  • Given the small cohort, statistical power is limited. Confidence intervals should accompany p-values, and nonparametric or bootstrap methods are preferable. Exploratory multivariate analyses (e.g., Cox regression) could identify independent prognostic factors.

We have updated our statistical methodology to use the non-parametric permutation test in place of the t-test used before. We also utilize bootstrapping to estimate 95% confidence intervals for the mean difference. This change is reflected in the relevant parts of the manuscript, however the statistical significance of the findings remain unchanged.

We have also conducted a multivariate Cox regression for the PFS (progression-free survival) to supplement Figure 5. We have normalized Age and KPS values to fall within [0,1] and used One-Hot-Encoding on the histology findings, but we grouped the smallest classes into an “Other” category to avoid overly specific (1-2 individuals for a single pathology) covariates. The coefficients and hazard ratios, with 95% confidence intervals along with p-values are shown below. We’d like to note, that none of these characteristics are sufficient for the prediction of PFS, and the small sample size results in wide confidence intervals. Figure 5 provides a visual summary of observed outcomes; however, given the lack of statistical significance and wide confidence intervals in the Cox regression, stratification by any single variable would not be justified and could be misleading.

coef [95% CI]

HR [95% CI]

p

Age (at surgery)

-0.844 [-41.634 - 39.947]

0.430 [>0.001 - >99,999]

0.968

Gender

0.136 [-4.325 - 4.596]

1.145 [0.013 - 99.10]

0.952

Intracranial Pressure Increase

-0.546 [-11.432 - 10.340]

0.579 [>0.001 - 30949]

0.922

Local Symptoms

0.297 [-5.055 - 5.650]

1.346 [0.006 - 284]

0.913

NPH/Chronic Hydrocephalus Symptoms

-1.497 [-13.174 - 10.180]

0.224 [>0.001 - 26380]

0.802

Symptom Onset

0.740 [-4.186 - 5.665]

2.095 [0.015 - 289]

0.768

Clinical Improvement

-0.434 [-10.322 - 9.453]

0.648 [>0.001 - 12752]

0.931

Stereotactic Radiosurgery

-0.140 [-3.340 - 3.059]

0.869 [0.035 - 21.31]

0.931

Conformal Radiotherapy

0.372 [-3.263 - 4.007]

1.450 [0.038 - 54.99]

0.841

Preop KPS

-2.272 [-45.255 - 40.712]

0.103 [>0.001 - >99,999]

0.918

Postop KPS

-2.230 [-16.482 - 12.023]

0.108 [>0.001 - >99,999]

0.759

Ambulatory Control KPS (First Post-Surgery)

-0.615 [-43.819 - 42.589]

0.541 [>0.001 - >99,999]

0.978

Ambulatory Control KPS (First Post-Radiotherapy)

-3.241 [-25.953 - 19.470]

0.039 [>0.001 - >99,999]

0.780

KPS at Last Ambulatory Visit

-0.571 [-11.645 - 10.503]

0.565 [>0.001 - 36439]

0.920

Histopathology:
Glioblastoma

1.371 [-17.969 - 20.710]

3.938 [>0.001 - >99,999]

0.890

Histopathology:
Papillary Tumor of the Pineal Region

1.759 [-13.315 - 16.833]

5.807 [>0.001 - >99,999]

0.819

Histopathology:
Pineal Parenchymal Tumor of Intermediate Differentiation

2.588 [-13.362 - 18.538]

13.300 [>0.001 - >99,999]

0.750

Histopathology:
Pineocytoma

2.434 [-12.900 - 17.767]

11.401 [>0.001 - >99,999]

0.756

Histopathology:
Other

2.335 [-13.493 - 18.164]

10.333 [>0.001 - >99,999]

0.772

  • Detailed description of dose, fractionation, target delineation, and planning technique is needed for reproducibility. Reporting organ-at-risk constraints, image guidance, and toxicity (CTCAE grading) would strengthen evaluation of safety and efficacy.

In accordance with the reviewer’s request, we collected the required information from the institutions where radiotherapy was administered and supplemented the manuscript with a new Table 1. We hope that all necessary information has now been fully provided.

  • The analysis should align with the WHO 2021 CNS classification and incorporate molecular markers (e.g., IDH mutation, MGMT, TERT). Including such data would enhance biological interpretation and comparability with current neuro-oncology standards.

In the newly added Table 1, we aimed to provide as detailed description of the histopathological characteristics of the tumors as possible. During our analyses, we refrained from applying a more detailed molecular classification, as stratification based on molecular markers would further reduce the interpretability of the statistical analyses given the low number of cases.

Reviewer 2 Report

Comments and Suggestions for Authors

Table 1 you have a column in which patients received neither SRS not RT, how were they treated

The patients received only RXT or a combination of RXT and CHT – you should mention that was the CHT protocol for cases receiving it

You have 2 meningioma cases – 1 with RXT but the other one?

What was the treatment for the sarcoma?

We agree that in selected cases this combination may be appropriate (biopsy + RXT/CHT), but there are certain pathologies where surgery is indicated as for the papillary tumor of the pineal region, pineoblastomas, teratomas and PPT with intermediate differentiation

SRS as standalone treatment is the current standard of care for pineocytomas and may be used in conjunction with CHT and RXT for GCT and pineoblastomas as an adjunct

It would be interesting to have the radiation doses

What was the number of patients that needed surgery? due to tumor progression after RxT

The authors should have a more open/flexible attitude on this complex and not quite frequent pathology, and not exclude the role of surgery for this pathology, for certain selected cases and histology this approach might be appropriate but definitely not for all cases

Author Response

We would like to thank the reviewer for their work, valuable comments, and constructive guidance.

  • Table 1 you have a column in which patients received neither SRS not RT, how were they treated

Reviewer no,1's request prompted a reevaluation of the radiotherapy issue, including detailed data collection from the institutions performing the radiotherapy. Please refer to the new Table 1 for the detailed results. Based on our final information, 5 patients did not receive treatment or dropped out of follow-up, due to poor clinical condition.

  • The patients received only RXT or a combination of RXT and CHT – you should mention that was the CHT protocol for cases receiving it

Patients received chemotherapy based on the histological type of the primary tumor under oncological supervision. Concomitant radio-chemotherapy was administered in two cases within our patient cohort: one patient with glioblastoma (patient ID: 26) and one with diffuse midline glioma (patient ID: 17).

  • You have 2 meningioma cases – 1 with RXT but the other one?

We have added this information in the new Table 1. Both patients with meningioma received fractionated stereotactic radiotherapy (fSRT).

  • What was the treatment for the sarcoma?

Due to a low Karnofsky performance status, the patient was not eligible for further oncotherapy, and follow-up was discontinued.

  • We agree that in selected cases this combination may be appropriate (biopsy + RXT/CHT), but there are certain pathologies where surgery is indicated as for the papillary tumor of the pineal region, pineoblastomas, teratomas and PPT with intermediate differentiation

In our cohort, there were no patients with pineoblastoma or teratoma. The main focus of the manuscript is to present the clinical practice at our department, as illustrated in Figure 6. If clinical deterioration is explained by triventricular hydrocephalus and sufficient improvement is observed following ETV, we generally initiate radiotherapy first for both papillary tumors of the pineal region and pineal parenchymal tumors. We have considerable experience with this approach.

We discuss the main conditions where surgery is indicated in our “Therapeutic algorithm recommendation”. As depicted in the flowchart (Figure 6), in cases of persistently poor condition after ETV, progression observed during follow-up, or tumor growth despite radiotherapy, direct surgical intervention is recommended at a specialized center by an experienced neurosurgeon.

  • SRS as standalone treatment is the current standard of care for pineocytomas and may be used in conjunction with CHT and RXT for GCT and pineoblastomas as an adjunct

We agree with the reviewer’s opinion.

  • It would be interesting to have the radiation doses

We added a new table. The new Table 1 contains the requested data.

  • What was the number of patients that needed surgery? due to tumor progression after RxT

As shown in Figure 4, direct surgical intervention was required in total four cases following the endoscopic procedure, two cases after the radiotherapy.

  • The authors should have a more open/flexible attitude on this complex and not quite frequent pathology, and not exclude the role of surgery for this pathology, for certain selected cases and histology this approach might be appropriate but definitely not for all cases

We fully concur with the reviewer's insightful perspective. The manuscript acknowledges the potential role of direct surgical intervention (it can be seen in the abstract, discussion, conclusion sections and in Figure 6 (Therapeutic algorithm) as well). Our clinical practice, as presented, prioritizes radiotherapy following the management of hydrocephalus and in patients exhibiting a satisfactory clinical state, owing to its more favorable side-effect profile. Nonetheless, surgical tumor resection remains an essential option and is judiciously employed when clinically indicated.

Round 2

Reviewer 1 Report

Comments and Suggestions for Authors

The revised manuscript has improved.

Reviewer 2 Report

Comments and Suggestions for Authors

Table 1 is a very nice improvement of your presentation, it would be nice to add in the same table CHT regimen used for treatment,

I imagine that metastatic lesions were not treated with standalone RxT, nor the GBMs

You have cases 6, 13, 16, 23 and 27 that are empty – it means that you have no data or they did not received any Rxt?

In table 2

You have to explain how it is possible for high grade gliomas to have a PFS of 14 days and a median OS of 1826 days ??
